# Rehabilitation Nutrition in Patients with Chronic Kidney Disease and Cachexia

**DOI:** 10.3390/nu14224722

**Published:** 2022-11-09

**Authors:** Masatsugu Okamura, Tatsuro Inoue, Masato Ogawa, Kengo Shirado, Nobuyuki Shirai, Takuma Yagi, Ryo Momosaki, Yoji Kokura

**Affiliations:** 1Berlin Institute of Health Center for Regenerative Therapies (BCRT), Charité–Universitätsmedizin Berlin, 13353 Berlin, Germany; 2Department of Rehabilitation Medicine, Yokohama City University Graduate School of Medicine, Yokohama 236-0004, Japan; 3Change Nutrition from Rehabilitation–Virtual Laboratory (CNR), Niigata 950-3198, Japan; 4Department of Physical Therapy, Niigata University of Health and Welfare, Niigata 950-3198, Japan; 5Division of Rehabilitation Medicine, Kobe University Hospital, Kobe 650-0017, Japan; 6Department of Rehabilitation, Aso Iizuka Hospital, Fukuoka 820-8505, Japan; 7Department of Rehabilitation, Niigata Rinko Hospital, Niigata 950-8725, Japan; 8Department of Rehabilitation, Hattori Hospital, Miki 673-0413, Japan; 9Department of Rehabilitation Medicine, Mie University Graduate School of Medicine, Tsu 514-8507, Japan; 10Department of Nutritional Management, Keiju Hatogaoka Integrated Facility for Medical and Long-Term Care, Hoso 927-0023, Japan

**Keywords:** rehabilitation nutrition, chronic kidney disease, cachexia, protein-energy wasting, nutritional disorders, exercise therapy

## Abstract

Rehabilitation nutrition is a proposed intervention strategy to improve nutritional status and physical function. However, rehabilitation nutrition in patients with cachexia and protein-energy wasting (PEW), which are the main nutrition-related problems in patients with chronic kidney disease (CKD), has not been fully clarified. Therefore, this review aimed to summarize the current evidence and interventions related to rehabilitation nutrition for cachexia and PEW in patients with CKD. CKD is a serious condition worldwide, with a significant impact on patient prognosis. In addition, CKD is easily complicated by nutrition-related problems such as cachexia and PEW owing to disease background- and treatment-related factors, which can further worsen the prognosis. Although nutritional management and exercise therapy are reportedly effective for cachexia and PEW, the effectiveness of combined nutrition and exercise interventions is less clear. In the future, rehabilitation nutrition addressing the nutritional problems associated with CKD will become more widespread as more scientific evidence accumulates. In clinical practice, early intervention in patients with CKD involving both nutrition and exercise after appropriate assessment may be necessary to improve patient outcomes.

## 1. Introduction

Chronic kidney disease (CKD) is a serious disease worldwide with a significant impact on patient prognosis [1]. CKD is known to be easily complicated by nutrition-related problems such as cachexia and protein-energy wasting (PEW) due to disease-background- and treatment- related factors [2,3], and evidence is mounting that these nutritional complications further worsen the prognosis of patients with CKD [4].

Rehabilitation nutrition has emerged as a strategy in recent years to address issues related to nutrition and physical function [5]. This strategy involves assessing the patient’s condition, setting clear goals, and implementing nutritional management and rehabilitation simultaneously. Recently, numerous research reports, guidelines, and position papers have been published on rehabilitation nutrition, and scientific evidence continues to accumulate [6,7].

Further studies have continued to clarify the effects of nutritional and exercise therapy alone and the relationship between nutritional status and exercise for cachexia and PEW in patients with CKD [8,9]. However, rehabilitation nutrition that combines nutritional and exercise therapy has not been extensively studied.

In this review, we aimed to summarize the reality of cachexia and PEW related to CKD, as well as the effectiveness of combined nutrition and exercise interventions, to provide strategies that can be applied in clinical practice.

## 2. Overview of Chronic Kidney Disease

### 2.1. Definition, Prevalence, and Prognosis of Chronic Kidney Disease

CKD is well-known as a severe and frightening public health issue worldwide [1]. CKD is a clinical condition secondary to changes in kidney function or structure, characterized by an irreversible, slow, and progressive course [10]. The major causes of CKD include diabetes, hypertension, chronic glomerulonephritis, chronic pyelonephritis, chronic use of anti-inflammatory medication, autoimmune diseases, polycystic kidney disease, congenital malformations, and prolonged acute kidney disease [10]. These progressive kidney diseases cause a gradual decline in kidney function over years to decades. CKD is diagnosed in adults when they present, for a period equal to or greater than three months, a glomerular filtration rate (GFR) lower than 60 mL/min/1.73 m^2^, or a GFR greater than 60 mL/min/1.73 m^2^, but with evidence of injury of the kidney structure [11]. As CKD progresses and approaches end-stage kidney disease, it eventually leads to the need for renal replacement therapy [12].

CKD has a high prevalence globally and has a significant impact on the medium- to long-term prognosis of patients. The overall estimated global prevalence of CKD stages 1–5 was reportedly 13.4%, and 10.6% for stages 3–5 [13]. In addition to its high prevalence, CKD is associated with a high risk of cardiovascular disease, critical illness, and death [14]. In 2013, global data reported that a lower estimated glomerular filtration rate (eGFR) was associated with 4% of global deaths, or 2.2 million deaths [15]. More than half of these deaths are attributed to cardiovascular causes, and 0.96 million deaths are associated with end-stage kidney disease [15]. In addition, the annual total mortality rate for patients on dialysis was reportedly 19.9%.

### 2.2. Physical Function, Muscle Strength, and Skeletal Muscle Mass in Patients with Chronic Kidney Disease

Patients with CKD have impaired physical function and reduced skeletal muscle mass, which have a significant impact on the patient’s prognosis. Physical function is impaired in patients with pre-dialysis CKD, with walking speed and 6-min walking distance reduced to 70% of that of normal participants and a prolonged Timed Up and Go test time of up to 40% of that of normal participants [16]. Walking speed and Timed Up and Go test time are more accurate predictors of mortality at three years compared with kidney function or serum biomarkers [16]. Among the risk factors for dialysis induction and death in patients with pre-dialysis CKD, weight loss reportedly had the highest impact with a 3.2 2-times increased risk, followed by 2.5-, 2.1-, 1.8-, and 1.7-times increased risk for frailty, decreased physical activity, walking speed, and grip strength, respectively [17]. Patients on dialysis have an even more marked decline in physical function, with peak oxygen uptake and walking speed up to approximately 60% lower than those in healthy individuals of the same age. In a previous study that stratified survival in patients on dialysis by muscle mass loss, reduced grip strength, or sarcopenia, mortality at four years was 1.35, 2.82, and 2.94 times worse in the muscle mass loss, reduced grip strength, and sarcopenia groups, respectively, compared with that in patients with adequate muscle mass and grip strength [18].

Patients with CKD often have reduced physical function and skeletal muscle mass, which plays a significant role in their prognosis. Therefore, implementing the appropriate nutritional and exercise interventions for skeletal muscle and physical function is important.

## 3. Overview of Cachexia

### 3.1. Definition, Prevalence, and Prognosis of Cachexia

Cachexia is a serious condition characterized by the weight and skeletal muscle mass loss that negatively impacts a patient’s quality of life (QOL) and prognosis [19]. Cachexia is caused by cancer, chronic heart failure, CKD, chronic obstructive pulmonary disease, autoimmune diseases, chronic infections, and sepsis. Cachexia was defined at a consensus conference held in Washington, DC, USA in 2006 by the following description: ‘cachexia, is a complex metabolic syndrome associated with underlying illness and characterized by loss of muscle with or without loss of fat mass’. The prominent clinical feature of cachexia is weight loss in adults and growth failure in children [19]. Currently, the widely accepted diagnostic criteria for cachexia are weight loss of at least 5% in 12 months or less plus three or more of the following five criteria: decreased muscle strength, fatigue, anorexia, low fat-free mass index, and abnormal biochemistry (Table 1). Various molecular mechanisms are involved in the development of cachexia in skeletal muscle, adipose tissue, digestive organs, central nervous systems, and immune systems. Humoral factors released from the tumour and immune responses and metabolic changes induced by the tumour are reportedly involved in the pathogenesis of cachexia, especially cancer-related cachexia [20,21,22,23].

Cachexia occurs more frequently in chronically ill patients and has a significant impact on their prognosis. More than 50% of cancer-related deaths are estimated to be complicated by cachexia, and cachexia is responsible for 20% of deaths among patients with cancer [24]. Cachexia is primarily associated with refractory disease and is more common in extremely severe conditions such as terminal illness. Thus, some reports suggest that cancer mortality rates are equal to the upper limit of patients affected by cachexia [25,26]. In patients with chronic heart failure, an estimated 10% minimum of outpatients with heart failure have cachexia [27,28].

### 3.2. Interventions of Cachexia

Recent clinical studies and basic experiments have indicated that cachexia is not an inevitable condition [29,30]. Cachexia, along with systemic inflammation, involves multiple factors, including anorexia, weight and skeletal muscle loss, and increased catabolism. Treatment and prevention of cachexia require multidisciplinary interventions that include pharmacotherapy as well as nutritional and exercise therapy, and psychological and social interventions [31]. In this combined approach, nutritional therapy is an important intervention that must be applied as early as possible to prevent systemic wasting, especially in the elderly (65 years and older) and chronically ill patients [32]. Increased protein intake up to 1.0–1.5 g/kg of body weight per day is recommended to prevent skeletal muscle wasting in older persons [33]. However, when limited to cancer-related cachexia, non-pharmacologic nutritional and exercise therapies present many challenges, including adherence. The few published randomized controlled trials did not demonstrate consistent benefits in outcomes such as weight, muscle strength, and QOL [34]. Clinical trials of multidisciplinary treatment combining nutritional therapy and exercise therapy are currently underway in Japan and overseas, and further developments are expected. [35,36]. Regarding drug therapy, anamorelin, a ghrelin agonist, was approved for manufacture and marketing in Japan in 2021 to increase body weight, muscle mass, and appetite in patients with cancer-related cachexia. [37].

Cachexia is associated with many factors that significantly worsen patient prognosis. In the future, developing multidisciplinary treatments that combine several interventions is desirable, rather than considering pharmacological and non-pharmacological therapies independently.

## 4. Chronic Kidney Disease and Cachexia/Protein-Energy Wasting

### 4.1. Previous Studies of Chronic Kidney Disease and Cachexia

CKD has been cited as one of the causes of cachexia [19], although surprisingly, the association between kidney failure and cachexia has been less clear. von Haehling et al. estimated the prevalence of CKD to be 0.1% of the population, with 50% of these patients having cachexia [2]. In addition, they estimated that 190,000 patients in Europe, 80,000 in the United States, and 30,000 in Japan have CKD and cachexia, with a mortality rate of 20% within one year. McKeaveney et al. examined the prevalence and clinical outcomes of cachexia in 106 patients with CKD on haemodialysis for the first time using the definition by Evans et al. [19]. The authors reported that 17 (16%) patients had cachexia, and that the cachexia group had significantly higher dialysis efficiency (urea reduction ratio) and significantly lower Functional Assessment of Anorexia/Cachexia Therapy scores, which examines appetite, than the non-cachexia group. In contrast, no significant differences were observed in body weight, body mass index, body composition, grip strength, albumin level and other blood test results, fatigue, or QOL [38]. Further studies on patients with CKD using a strict definition of cachexia would be desirable in the future.

### 4.2. Definition, Prevalence, and Prognosis of Protein-Energy Wasting

Several studies have examined PEW, a condition similar to cachexia, in patients with CKD. PEW as a concept was proposed by the International Society of Renal Nutrition and Metabolism (ISRNM) and the International Society of Nephrology in 2008, and is defined as the state of decreased body stores of protein and energy fuels (i.e., body protein and fat masses) [3]. The diagnostic criteria for PEW are three or more of the following four criteria, which are similar in many ways, if not identical, to cachexia: low biochemical criteria, low body weight, decreased muscle mass, and low protein and energy intake (Table 1). Koppe et al. suggest that cachexia is a more advanced and severe form of PEW [39].

PEW in patients with CKD is directly related to nutritional factors and has a very poor prognosis. The causes of PEW include anorexia, decreased energy and protein intake, hypermetabolism, uraemia, metabolic acidosis, decreased physical activity, decreased anabolism, comorbidities (diabetes, chronic heart failure, coronary artery disease, peripheral artery disease, depression), and dialysis [3,40] (Figure 1). Previous studies have reported that PEW is more likely to occur in patients with CKD stage 3b (eGFR < 45 mL/min) as defined by the Kidney Disease Improving Global Outcomes CKD staging [41]. Carrero et al. found that 60–82% of patients with acute kidney disease, 11–54% of patients with CKD stages 3–5, and 28–52% of renal transplant patients have concomitant PEW [42]. Shirai et al. reported that nutrition-related problems in patients with CKD lead to decreased physical activity, declined muscle weakness, and falls, which further worsen the prognosis [43]. de Mutsert et al. reported that in dialysis patients with PEW defined by the subjective global assessment of nutritional status, the risk of death at 7 years is doubled, and 5 times higher in severe PEW at 6 months [4]. In patients with CKD, cachexia, which is considered a more advanced and severe form of PEW [39], is expected to have an even worse mid- to long-term prognosis than PEW. Thus, because of the high prevalence of cachexia and PEW in patients with CKD and their significant impact on the patient’s prognosis, establishing interventions for these conditions is important.

Patients with CKD can easily develop cachexia and PEW owing to the pathophysiology of the disease and treatments such as dialysis, and these conditions can worsen the mid- to long-term prognosis. Multiple factors, such as poor nutrition and skeletal muscle loss, are involved in cachexia and PEW, and a multidisciplinary approach based on nutritional and exercise therapies is important.

## 5. Rehabilitation Nutrition in Chronic Kidney Disease and Cachexia/Protein-Energy Wasting

### 5.1. Rehabilitation Nutrition

Nutritional management and exercise therapy based on the concept of rehabilitation nutrition are important for the treatment of cachexia and PEW in patients with CKD. Rehabilitation nutrition refers to nutritional management based on the International Classification of Functioning, Disability and Health (ICF), including nutritional status, to maximize the function, activity and participation of disabled and elderly people [5,44]. Rehabilitation nutrition is also defined as: (i) evaluating patients holistically using the ICF to assess causes of nutritional disorders, sarcopenia, and excessive or deficient nutritional intake; (ii) conducting rehabilitation nutrition diagnosis and goal setting; (iii) using ‘nutrition care management with rehabilitation in mind’ and ‘nutritionally conscious rehabilitation’ for patients who are disabled and frail older patients; (iv) improving nutritional status, sarcopenia, and frailty; and (v) promoting optimal physical function, activity, participation, and QOL [6,7]. In sarcopenia, a loss of skeletal muscle mass and strength, a combination of nutritional and exercise therapy is more effective than each intervention alone. The Asian Working Group for Sarcopenia 2019 (AWGS2019), a working group on sarcopenia in Asia, showed that a combination of nutritional and exercise therapy improves muscle strength and function [45]. The European Working Group on Sarcopenia in Older People 2 (EWGSOP2), a working group on sarcopenia in Europe, stated that considering a combination of protein intake and exercise is important, as well as an active lifestyle [46].

### 5.2. Nutritional Management

Patients with CKD are prone to undernutrition because of decreased protein and energy stores, and aggressive nutritional management is recommended [8]. The ISRNM issued a consensus statement on nutritional interventions for the prevention and treatment of PEW in patients with CKD [47]. The following conditions are considered indicators of nutritional intervention: anorexia or poor oral intake; protein intake < 1.2 g/kg/day (CKD stage 5D) or <0.7 g/kg/day (CKD stages 3–4); energy intake < 30 kcal/kg/day; serum albumin < 3.8 g/dL or serum pre-albumin level < 28 mg/dL; unintentional weight loss over 3 months > 5% of ideal weight or estimated dry weight; worsening nutritional markers over time; and subjective global assessment reaching the PEW range. Regular monitoring and assessment of patients with CKD for cachexia and proactive implementation of interventions are important [12]. In patients who need dietary support, nutrition interventions using behavioural counselling to promote the patient’s awareness and self-management of their condition are reportedly effective in the nutritional management of CKD [48]. When dietary counselling and standard preventive interventions fail to achieve the above nutritional intervention requirements, dietary supplements are recommended as nutritional support for patients with CKD [49]. When dietary intake and oral nutritional supplements are not sufficient to maintain adequate nutritional status, tube feeding or intravenous nutrition may be necessary and should be considered.

Some recent studies have shown positive results for the administration of proteins to patients with CKD. Guidelines have recommended limiting protein intake for patients with CKD. The US guidelines recommend a protein intake of 0.55–0.6 g/kg/day for patients with CKD stages 3–5 (except for stage 5D) without diabetes to reduce the risk of transition to end-stage kidney disease [50,51]. In addition, a protein intake of 0.6–0.8 g/kg/day is recommended for patients with CKD stages 3–5 (except for stage 5D) with diabetes to maintain nutritional status and glycaemic control [50,51]. In contrast, in elderly patients with CKD, high protein intake is not associated with reduced kidney function, although the low-protein-intake group reportedly lost a high degree of body weight in one year [52]. High protein intake is reportedly not associated with the risk of developing end-stage kidney disease in elderly patients with CKD [53]. In addition, plant proteins are more likely to inhibit kidney function decline than animal proteins [54]. Recently, a position paper was issued that recommends less restrictive and more aggressive nutritional therapy for patients with CKD depending on their condition [55]. Therefore, both protein restriction and promotion are necessary in patients with CKD, considering factors such as the patient’s pathology and age, rather than uniformly restricting protein in all patients with CKD.

Recently, β-hydroxy-β-methylabutyrate (HMB), L-carnitine, and branched-chain amino acids (BCAAs) have received attention in nutritional therapy for CKD. HMB is a metabolite of leucine, which does not affect kidney function in patients on haemodialysis [56], and can reportedly improve muscle mass and physical function in older people [57]. Patients on dialysis have carnitine deficiency, which is associated with poor physical function [58,59]. L-carnitine administration to patients on dialysis can reportedly improve QOL and exercise tolerance [60,61], as well as the accumulation of aging substances associated with prognosis, and reduce depression [62,63]. BCAAs can improve muscle structure and function in patients with sarcopenia [64]. Because many BCAAs are lost during dialysis, previous studies have reported that blood BCAA levels are maintained in patients on dialysis using dialysate containing physiological concentrations of BCAAs or consuming protein before dialysis [65,66].

Nutritional therapy for patients with CKD may need to consider the patient’s age and condition concerning protein intake. Further evidence should be accumulated regarding protein intake, including other nutrients.

### 5.3. Exercise Therapy

Patients with CKD are prone to reduced skeletal muscle function and exercise tolerance, and exercise therapy based on the patient’s condition is an essential intervention. Clinical practice guidelines on renal rehabilitation recommend exercise therapy to improve outcomes such as exercise tolerance and QOL in patients with both non-dialysis and dialysis [67]. Renal rehabilitation is also associated with reduced mortality [68]. The ISRNM consensus statement on the prevention and treatment of PEW in patients with CKD categorizes exercise therapy as an adjunctive therapy. However, systematic reviews of exercise therapy for patients with CKD reported improvements in skeletal muscle mass, muscle strength, exercise tolerance, walking ability, cardiac function, nutritional indicators, and health-related QOL [9,69,70]. In particular, incremental resistance training is useful for increasing skeletal muscle mass, muscle strength, and health-related QOL, which are subcategories of the PEW diagnostic criteria. [71,72]. Recently, muscle power, defined by muscle strength and muscle contraction rate, was reportedly reduced in patients with CKD [73]. Therefore, to improve physical function in patients with CKD, incorporating training that focuses on movement speed may have better results than training to increase muscle mass [73].

In patients with CKD, increasing the amount of usual physical activity is important in addition to exercise therapy. Previous observational studies have reported an association between physical activity and mortality. In a large cohort study of 20,920 patients on haemodialysis, those who were physically active at least once a week had higher survival rates than inactive patients [74]. Patients who were sedentary at the start of dialysis treatment had a 62% increased risk of death compared with active patients [75]. A large study in community-dwelling older adults found that physical activity and exercise interventions reduce the decline in the eGFR [76]. Patients who walk more frequently have a lower 10-year risk of death and a lower rate of transition to renal replacement therapy [77]. Patients with higher rates of participation in programs to improve physical activity also have lower rates of cardiovascular events and mortality [78].

Exercise therapy for patients with CKD must consider several risks. Patients with CKD often have conditions that put them at risk for exercise therapy, such as cardiovascular disease, as well as cachexia and PEW. Therefore, starting with low-intensity, low-frequency, and short-duration exercise therapy is important at the beginning of exercise therapy, with gradual increases to ensure safety. Incremental resistance training during haemodialysis can improve PEW scores in addition to improving outcomes such as physical function and skeletal muscle mass loss [79]. Exercise therapy was previously considered a contraindication for patients with CKD. However, the previous study has shown that kidney-related parameters do not change before or after exercise corresponding to four metabolic equivalents of task [80]. Blood flow to the kidney is reportedly not reduced by up to the anaerobic threshold level of exercise in patients with CKD [81]. Clearly, exercise therapy can improve physical function without worsening kidney function [82].

### 5.4. Rehabilitation Nutrition for Chronic Kidney Disease

Rehabilitation nutrition for cachexia and PEW in CKD may be effective. To date, the effectiveness of combined nutritional management and exercise therapy interventions for cachexia and PEW in CKD has not been demonstrated. However, regarding kidney disease, a position paper from Japan recommends nutritional physiotherapy combined with aerobic exercise and resistance training to improve the patient’s function [83]. The exercise intensity of these interventions should be adjusted according to age and physical function [83]. Physical function can decline in patients with CKD as the CKD stage progresses [84]. Protein intake positively correlates with muscle strength in patients with CKD [85]. In recent years, the concept of uraemic sarcopenia has gained attention in patients with CKD [86]. Sarcopenia is characterized by decreased muscle mass, muscle strength, and physical performance, which is common to the subcategories of the diagnostic criteria for cachexia and PEW [45,46]. Therefore, sarcopenia, cachexia, and PEW are thought to have similar mechanisms and bidirectional relationships [41]. Rehabilitation nutrition interventions are effective for sarcopenia [87,88]. Therefore, rehabilitation nutrition combined with assessment-based nutritional management and exercise therapy may be effective for patients with CKD who present with nutrition-related problems similar to sarcopenia, such as cachexia and PEW.

Rehabilitation nutrition for CKD requires aggressive interventions based on regular assessment. Figure 2 shows a flowchart of our proposed rehabilitation nutrition strategy for cachexia and PEW in patients with CKD. Nutritional disorders are common complications in patients with CKD, and screening and assessing patients for cachexia, PEW, and nutritional status regularly every month is important [71,89]. In patients on dialysis, weight fluctuations between treatments should also be considered. Conducting a multidimensional and comprehensive assessment of diet and nutritional intake, nutritional status, physical function, and physical activity is important in daily clinical practice, while also referring to the ISRNM nutrition intervention indicators presented earlier. [47]. The evaluation should be conducted holistically using ICF. After the assessment, nutritional disorders including cachexia and PEW, as well as their causes, should be properly diagnosed, and SMART (specific, measurable, achievable, relevant, and time-bound) goals should be set according to the patient’s condition. Implementation of combined nutrition and exercise therapy interventions is important, and if no improvement is shown, a different approach should be applied and re-evaluated. In addition, advanced nutritional management may be more beneficial than imposing protein restrictions, and should be considered based on the patient’s condition [55].

Further evidence is needed on rehabilitation nutrition for cachexia and PEW in patients with CKD. Currently, no treatment fundamentally improves cachexia in patients with CKD. The effects of nutritional and exercise interventions alone in patients with CKD are gradually becoming clear. However, the effects of rehabilitation nutrition on cachexia and PEW are not clear. In the future, accumulating further scientific evidence as well as promotion of early detection and intervention for cachexia and PEW using multidimensional patient evaluations is necessary.

## 6. Other Interventions

Several pharmacological interventions have been developed for cachexia and PEW in patients with CKD. Clearly, anabolic steroids (e.g., testosterone, nandrolone decanoate) improve body weight, lean body mass, and muscle mass in humans [90,91], and recombinant human growth hormone improves nutritional markers [92]. Anabolic steroids also have side effects such as elevated transaminase concentrations, decreased high-density lipoprotein concentrations, interaction with oral anticoagulants or oral hypoglycaemics, and hypogonadism [90]. In animal experiments, ghrelin and synthetic ghrelin receptor agonists improved appetite, lean body mass, inflammation, and muscle anabolism [93]. These drugs are still in the research and development stage, and future studies are required to verify the effectiveness of these drugs in combination with nutritional and exercise interventions.

## 7. Conclusions

Rehabilitation nutrition for cachexia and PEW in patients with CKD may be effective to improve outcomes such as exercise tolerance, physical function, and QOL. In future studies, clarification of the actual detailed status of cachexia and PEW in patients with CKD and the accumulation of scientific evidence on rehabilitation nutrition combined with nutritional management and exercise therapy is desirable.

## Figures and Tables

**Figure 1 nutrients-14-04722-f001:**
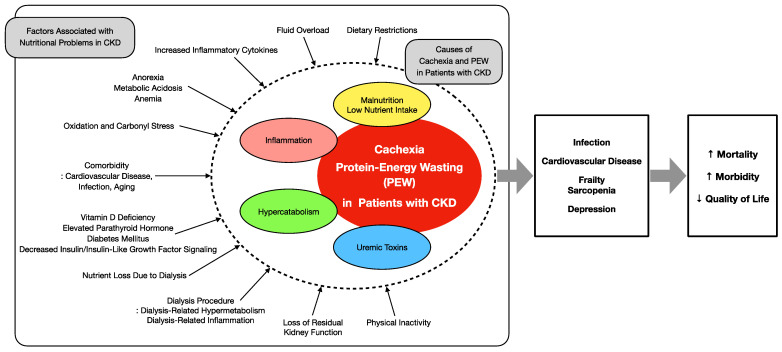
Conceptual diagram of cachexia and protein-energy wasting in patients with chronic kidney disease. CKD, chronic kidney disease; PEW, protein-energy wasting.

**Figure 2 nutrients-14-04722-f002:**
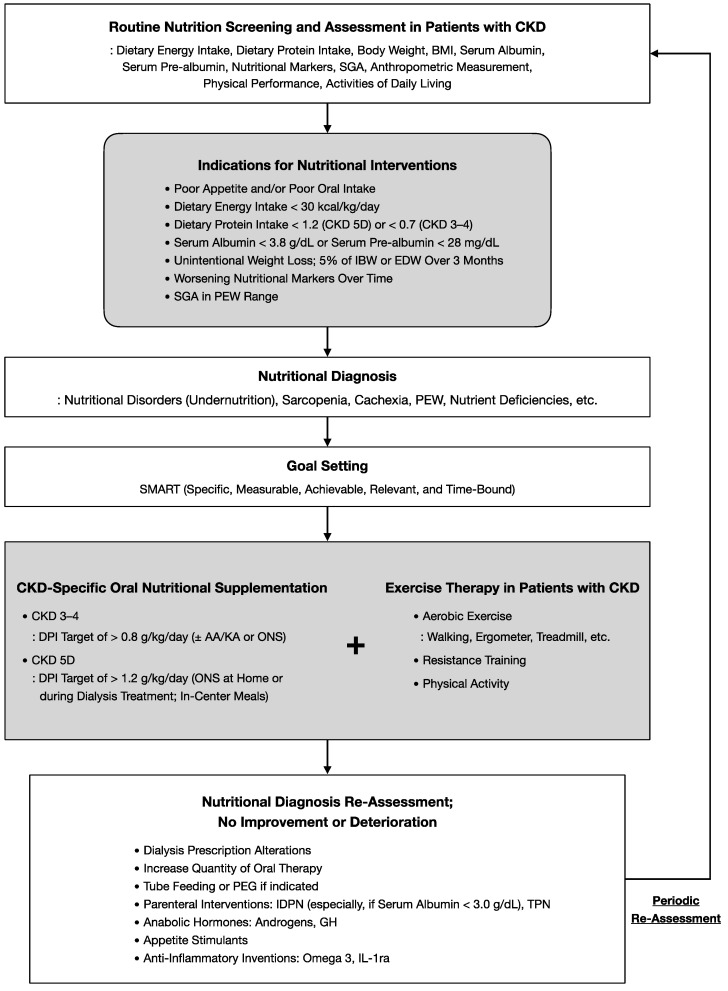
Flowchart of rehabilitation nutrition in patients with chronic kidney disease. AA/KA, amino acid/keto acid; BMI, body mass index; CKD, chronic kidney disease; DPI, dietary protein intake; EDW, estimated dry weight; GH, growth hormone; IBW, ideal body weight; IDPN, intradialytic parenteral nutrition; IL-1ra, interleukin-1 receptor antagonist; ONS, oral nutritional supplement; PEG, percutaneous endoscopic gastrostomy; PEW, protein-energy wasting; SGA, subjective global assessment; TPN, total parenteral nutrition, adapted from [5,47,83].

**Table 1 nutrients-14-04722-t001:** Criteria for the clinical diagnosis of cachexia and protein-energy wasting.

Criteria	Cachexia (Cachexia Consensus Working Group)	Protein-Energy Wasting (International Society of Renal Nutrition and Metabolism)
Dietary intake (Anorexia)	Unintentional low dietary energy intake < 20 kcal/kg/day Unintentional low dietary energy intake < 70% of usual food intake Poor appetite	Unintentional low dietary protein intake < 0.80 g/kg/day for at least 2 months for dialysis patients or <0.6 g/kg/day for patients with CKD stages 2–5 Unintentional low dietary energy intake < 25 kcal/kg/day for at least 2 months
Serum chemistry	Serum albumin < 3.2 g/dL	Serum albumin < 3.8 g/dL
	Anaemia: haemoglobin < 12 g/dL	Serum pre-albumin (transthyretin) < 30 mg/dL (for maintenance dialysis patients only; levels may vary according to GFR level for patients with CKD stages 2–5)
	Increased inflammatory markers: CRP > 0.5 mg/dL, IL-6 > 4.6 pg/mL	Serum cholesterol < 100 mg/dL
Body mass	N/A	BMI < 23 kg/m^2^ (a lower BMI might be desirable for certain Asian population; weight must be oedema-free mass, for example, post-dialysis dry weight) Unintentional weight loss over time: 5% over 3 months or 10% over 6 months Total body fat percentage < 10%
Muscle mass	Reduced mid-upper arm muscle circumference < 10th percentile for age and sex Reduction in appendicle skeletal muscle index on DEXA (kg/m^2^) by <5.45 and <7.25 in women and men, respectively	Muscle wasting: reduced muscle mass 5% over 3 months or 10% over 6 months Reduced mid-arm muscle circumference (reduction > 10% in relation to 50th percentile of reference population) Low creatinine appearance
Muscle strength	Decreased muscle strength (lowest tertile, e.g., handgrip strength	N/A
Fatigue	Physical or mental weariness resulting from exertion; inability to continue exercise at the same intensity with a resultant deterioration in performance	N/A
Diagnosis of cachexia/PEW	Weight loss of at least 5% in 12 months or less in the presence of underlying illness, plus three of the other criteria	At least three of the four listed categories (and at least one test in each of the selected categories)

BMI, body mass index; CKD, chronic kidney disease; CRP, C-reactive protein; DEXA, dual-energy X-ray absorptiometry; GFR, glomerular filtration rate; IL-6, interleukin-6; N/A, not applicable; PEW, protein-energy wasting.

## Data Availability

Not applicable.

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
