# Peer review of "Rehabilitation Nutrition in Patients with Chronic Kidney Disease and Cachexia"

_nutrients, 2022, doi:10.3390/nu14224722_

Round 1

Author Response

Response to Reviewer 1 Comments

We wish to express our appreciation to the reviewer for his/her insightful comments, which have helped us significantly improve the paper.

Point 1: At first, I have issues with the way they describe this review, and to me, use of some tables, such as-a PRISMA diagram should be considered while selecting papers that described the significant impact of dietary intake and physical exercise on the physiological well-being of CKD patients, so that, at a quick glance, the reader can go through those papers.

Response 1: PRISMA (Preferred Reporting Items for Systematic Reviews and Meta-Analyses) diagram is a figure visualizing the flow from literature search to screening during a systematic review. As this manuscript is a narrative review, it does not deserve the use of PRISMA diagram. However, in this review, we have described the papers we incorporated after the search with the agreement of all authors.

Point 2: Although aerobic training could help in improving physical functioning of patients, it is not always practical and feasible for patients.

Response 2: According to the reviewer’s constructive comment, aerobic exercise may not be practical, but it may be possible to adapt the exercise intensity, including walking, to the patient’s condition. Therefore, we have added the following;

The exercise intensity of these interventions should be adjusted according to age and physical function [83]. (Line 349)

Point 3: Another thing that I would like to see in this manuscript was how we can help those unfortunate patients with dietary intake to regain their health status-but this was not clear.

Response 3: According to the reviewer’s constructive comment, we have described as follows;

In patients who need dietary support, nutrition interventions using behavioural counselling to promote the patient’s awareness and self-management of their condition are reportedly effective in the nutritional management of CKD [48]. When dietary counselling and standard preventive interventions fail to achieve the above nutritional intervention requirements, dietary supplements are recommended as nutritional support for patients with CKD [49]. When dietary intake and oral nutritional supplements are not sufficient to maintain adequate nutritional status, tube feeding or intravenous nutrition may be necessary and should be considered. (Line 247)

Point 4: Diagnosis of PEW and cachexia separately is also difficult as they possess almost similar features.

Response 4: According to the reviewer’s constructive comment, cachexia and PEW are similar conditions. We have already described the following with the previous literature in the text;

The diagnostic criteria for PEW are three or more of the following four criteria: low biochemical criteria, low body weight, decreased muscle mass, and low protein and energy intake, which are similar in many ways, if not identical, to cachexia (Table 1). Koppe et al. suggest that cachexia is a more advanced and severe form of PEW [39]. (Line 185)

Point 5: In page 9, line 361-369, the authors mentioned the use of anabolic hormones, but will there be any adverse effect due to those usage?

Response 5: According to the reviewer’s constructive comment, anabolic hormones have some side effects. Therefore, we have added the following text;

Anabolic steroids also have side effects such as elevated transaminase concentrations, decreased high-density lipoprotein concentrations, interaction with oral anticoagulants or oral hypoglycaemics, and hypogonadism [90]. (Line 374)

(Please see the attachment)

Reviewer 2 Report

Comment to the author

The review article is well written about rehabilitation nutrition in CKD and cachexia.  The article introduce the definition of CKD/cachexia,interaction of CKD and cachexia/PEW,intervention for cachexia/PEW in CKD including nutritional aspect and rehabilitation aspects in detail. I suggest paper accept after minor revision

Some minor error needed to be corrected.

Line 32,57: I suggest change "the effects of nutrition and exercise interventions combined" to "the effects of combined nutrition and exercise interventions"

Line 140: Protein intake up to 1.2 to 1.5g/kg is suggested in hemodialysis patients but not advanced CKD patients

Line 193:de Mutsert>De Mutsert

PS: Potential conflict of interest with regards to the paper exists since the corresponding author are identical to the special issue editor

Author Response

Response to Reviewer 2 Comments

We wish to express our appreciation to the reviewer for his/her insightful comments, which have helped us significantly improve the paper.

Point 1: Line 32,57: I suggest change "the effects of nutrition and exercise interventions combined" to "the effects of combined nutrition and exercise interventions"

Response 1: We have corrected to “the effectiveness of combined nutrition and exercise interventions” in these sentences. (Line 32 and 58)

Point 2: Line 140: Protein intake up to 1.2 to 1.5g/kg is suggested in hemodialysis patients but not advanced CKD patients

Response 2: We had incorrectly described this sentence. Therefore, we have corrected it as follows; Increased protein intake up to 1.0–1.5 g/kg of body weight per day is recommended to prevent skeletal muscle wasting in older persons. (Line 141)

Point 3: Line 193:de Mutsert>De Mutsert

Response 3: We have listed this author’s name (de Mutsert) according to the citation notation in PubMed. (https://pubmed.ncbi.nlm.nih.gov/19144733/) (Line 200)

Point 4: PS: Potential conflict of interest with regards to the paper exists since the corresponding author are identical to the special issue editor

Response 4: According to the reviewer’s constructive comment, the responsible author is also the special issue editor. Therefore, we have added the following to “Conflicts of Interest”;

The corresponding author (Y.K.) is identical to the special issue editor. A different editor is responsible for this review, and Y.K. was not involved in any decisions and the selection of the reviewers. Therefore, the responsible author (Y.K.) also has no conflict of interest. (Line 404)

(Please see the attachment)
